



# Measurement Report: Interpretation of Wide Range Particulate Matter Size Distributions in Delhi

Ülkü Alver Şahin[1], Roy M Harrison[2,a], Mohammed S. Alam[2,b]
David C.S. Beddows[2,3], Dimitrios Bousiotis[2], Zongbo Shi[2]
Leigh R. Crilley[4], William Bloss[2], James Brean[2], Isha Khanna[5,c]
and Rulan Verma[6,c]

[1] Istanbul University-Cerrahpaşa, Engineering Faculty Environmental
Engineering Department, Istanbul, Turkey

[2] School of Geography, Earth and Environmental Sciences
University of Birmingham, Birmingham, B15 2TT, UK

[3] National Centre for Atmospheric Science
University of York, Heslington, York, YO10 5DQ, UK

[4] Department of Chemistry, York University
Toronto, Ontario, M3J 1P3, Canada

[5] Puget Sound Clean Air Agency, Seattle, Washington, USA 98101

[6] Institute of research on catalysis and the environment of Lyon - IRCELYON
Université De Lyon, 69626 Villeurbanne cedex, France

Corresponding author:  E-mail: r.m.harrison@bham.ac.uk (Roy M. Harrison)

[a] Also at: Department of Environmental Sciences / Centre of Excellence in Environmental
Studies, King Abdulaziz University, PO Box 80203, Jeddah, 21589, Saudi Arabia

[b] Now at:  School of Biosciences, University of Nottingham, Sutton Bonington
Campus, Leicestershire, LE12 5RD

[c] Previously at: IIT Delhi, Hauz Khas, New Delhi, India 110016



**ABSTRACT**
Delhi is one of the world's most polluted cities, with very high concentrations of airborne
particulate matter. However, little is known on the factors controlling the characteristics of particle
number size distributions. Here, new measurements are reported from three field campaigns
conducted in winter, pre-monsoon and post-monsoon seasons on the Indian Insitute of Technology
campus in the south of the city. Particle number size distributions were measured simultaneously
using a Scanning Mobility Particle Sizer and a Grimm optical particle monitor, covering 15 nm to
>10 µm diameter. The merged, wide-range size distributions were categorised into five size ranges:
nucleation (15-20 nm), Aitken (20-100 nm), accumulation (100 nm-1 µm), large fine (1-2.5 µm)
and coarse (2.5-10 µm) particles. The ultrafine fraction (15-100 nm) accounts for about 52 % of all
particles by number ($PN_{10}$), but just 1 % by $PM_{10}$ volume ($PV_{10}$). The measured size distributions
are markedly coarser than most from other parts of the world, but are consistent with earlier cascade
impactor data from Delhi. Our results suggest substantial aerosol processing by coagulation,
condensation and water uptake in the heavily polluted atmosphere, which takes place mostly at
nighttime and in the morning hours. Total number concentrations are highest in winter, but the
mode of the distribution is largest in the post-monsoon (autumn) season. The accumulation mode
particles dominate the particle volume in autumn and winter, while the coarse mode dominates in
summer. Polar plots show a huge variation between both size fractions in the same season and
between seasons for the same size fraction. The diurnal pattern of particle numbers is strongly
reflective of a road traffic influence upon concentrations, especially in autumn and winter. There is
a clear influence of diesel traffic at nighttime when it is permitted to enter the city, and also
indications in the size distribution data of a mode <15 nm, probably attributable to CNG/LPG
vehicles. New particle formation appears to be infrequent, and in this dataset is limited to one day
in the summer campaign. Our results reveal that the very high emissions of airborne particles in
Delhi, particularly from traffic, determine the variation of particle number size distributions.



**1. INTRODUCTION**
Air pollution in Delhi has been studied for many years, and the authorities have implemented several
interventions designed to limit the concentrations. The sulphur content of diesel and petrol fuels was
reduced to 50 ppm during 1996-2010, more than 1300 industries were shut down due to hazardous
emissions, commercial vehicles older than 15 years were gradually taken out of the traffic fleet, and
public transport vehicles and auto-ricksaws were converted to compressed natural gas (CNG) fuel
(Narain and Krupnick, 2007). An odd–even vehicle number plate restriction has been applied during
working days (Chowdhury et al., 2017). Although these measures have reduced gaseous pollutants
($SO_2$ and CO) and primary particulate matter, in recent years, several studies have reported that the
$PM_{2.5}$ concentrations have been constant or slowly increasing in India, especially in the winter and
autumn seasons (Babu et al., 2013; Balakrishnan et al., 2019; Dandona et al. 2017, Kumar et al.,
2017), except in 2020. In 2020, the $PM_{2.5}$ level decreased by approximately 40 %, due to Covid-19
measures (Rodríguez-Urrego and Rodríguez-Urrego 2020; Mahato et al., 2020). Although the overall
emission sources in India are dominated by traffic, industry, construction, and local biomass burning,
haze pollution events in Delhi are frequently related to the large-scale open burning of post-harvest
crop residues/wood during the crop burning season in nearby rural regions (Cusworth et al. 2018;
Bikkina et al. 2019; Kanawade et al., 2020). Although the sources of particles are mostly local (Hama
et al., 2020), meteorological factors play an important role in influencing concentrations (Tiwari et
al., 2014; Yadav et al., 2016; Guo et al., 2017; Dumka et al. 2019; Kumar et al. 2020).

Annual average $PM_{2.5}$ levels range between 81 and 190 µg/m$^3$ in Delhi and are clearly higher than
the WHO guideline value (5 µg/m$^3$) and Indian national limit value (40 µg/m$^3$) (Hama et al., 2020).
To the best of our knowledge, most studies in India have focussed on the source apportionment from
chemical profiles of particles (Pant and Harrison, 2012; Jain et al. 2020; Bhandari et al., 2020; Rai
et al., 2020). Mostly they have reported that biomass burning contributes greatly to $PM_{2.5}$ while traffic
contributes heavily to $PM_{10}$ in Delhi. Residential energy use contributes 50 % of the $PM_{2.5}$



concentration and the construction sectors are also evaluated as an important source of particles
(Guttikunda et al., 2014; Butt et al., 2016; Conibear et al., 2018). Furthermore, it is particularly
important to understand the absolute contribution and sources of different sizes of particles within
$PM_{2.5}$. A recently published paper by Das et al. (2021) highlighted that <250 nm particles contribute
a significant proportion of the total $PM_{2.5}$ and are a potentially important link with human health.

The Particle Size Distribution (PSD) can provide air pollution source apportionment with high time
resolution compared to use of chemical species, and influences the aerosol transport and
transformation profiles in the urban atmosphere and toxicological effects on humans (Wu and Boor,
2021). Many PSD studies have been conducted in urban, traffic and background sites over the past
decades and three review studies have been published (Vu et al., 2015; Azimi et al., 2014; Wu and
Boor, 2021). There are some studies evaluating the number or mass PSD in Delhi (Mönkkönen et al.,
2005; Chelani et al., 2010; Gupta et al., 2011; Pant et al., 2016; Gani et al., 2020). Harrison (2020)
compared PSDs from Delhi, Beijing and London and reported that the particles from Delhi are far
greater in number with a much larger modal diameter, close to 100 nm. In a recent paper, Gani et al.
(2020) has investigated the PSD up to 0.5 µm sizes from 2017 to 2018 and reported that rapid
coagulation is an important process in Delhi.

The wide range PSD is important to describe all sources of inhalable particles (<10 µm). It is not easy
to separate particles arising from resuspension, sea salt and construction, or from brake wear and
combustion or vehicle exhaust, using only <0.5 µm particle sizes. Harrison et al. 2011 reported that
using wide range particle sizes in source apportionment is extremely successful in identifying the
separate contributions of on-road emission including brake wear and resuspension. Although there
are a few studies of wide range particle characterization in Beijing (Jing et al., 2014) and source
apportionment in Venice, Italy (Masiol et al., 2016), there has been no wide range PSD study in Delhi.
In this study, we aimed to interpret particulate matter size distributions over a wide range (15 nm to



10 µm) in the winter, post-monsoon and pre-monsoon seasons in Delhi. Future studies will look at
two-step receptor modelling of wide range particulate matter size distributions and chemical
composition in Delhi.

**2.    METHODS**
**2.1    Study Area**
The measurements were part of the NERC/MoES Air Pollution and Human Health in an Indian mega-
city (APHH-Delhi, www.urbanair-india.org) study, a joint UK-India project addressing air pollution
in Delhi. The sampling location was ~15 m above ground level on the 4th floor of the Civil
Engineering Department at the Indian Institute of Technology Delhi (IIT Delhi) campus, located in
New Delhi, representative of an urban background environment (28.545 N, 77.193 E) (Figure S1).
As part of APHH-Delhi, there were three field campaigns: (i) Jan-Feb 2018 (winter), (ii) May-June
2018 (summer; pre-monsoon) and (iii) Oct-Nov 2018 (autumn; post-monsoon). In all field
campaigns, a suite of gas and particulate phase instrumentation was deployed within a temperature
controlled laboratory.

These sampling periods were representative of conditions for PM and gases during these seasons in
Delhi. We found the average $PM_{2.5}$ concentration to be approximately 180 µg/m$^3$, 220 µg/m$^3$ and 120
µg/m$^3$ for winter, autumn (excluding Diwali) and summer, respectively measured by a TEOM-
FDMS. Hama et al. (2020) studied the long term (from 2014 to 2017) trends of air pollution in Delhi
at 6 stations (residential, commercial, and industrial sites) and reported that the mean $PM_{2.5}$
concentrations ranged between 147 – 248 µg/m$^3$, 147 – 248 µg/m$^3$ and 76 – 135 µg/m$^3$ for winter,
autumn and summer, respectively, and a good correlation between sites within Delhi. This gives
reassurance that the $PM_{2.5}$ concentrations measured at our site are within the typical range of those
observed in Delhi.





## 2.2 Measurements

To measure the particle size range used in this study, two particle instruments were used to collect number size distributions (NSD). For the range 15-640nm, a TSI SMPS 3936 was used, consisting of a TSI 3080 Electrostatic Classifier, TSI 3081 DMA and TSI 3775 CPC. To extend this range into the coarse mode a GRIMM 1.108 Portable Laser Aerosol Spectrometer and Dust Monitor were used alongside the SMPS.

Aerosol particle sizes in the atmosphere span a very wide range from a few nanometers at the lower end to some tens of micrometers at the upper end. Because of this very wide range of sizes, particle properties vary considerably across the size spectrum with the behaviour of the smaller particles being determined by their high mobility and hence diffusivity, whilst at the coarse end of the size distribution inertial properties are especially important. Due to this divergence in behaviour, no instrument is capable of measurement of the whole range of particle sizes. The smaller particles are mostly measured as a function of their electric mobility when charged, while the larger particles are counted using their inertial or optical properties. In this study an SMPS (Scanning mobility particle sizer) based on mobility diameters and a GRIMM optical spectrometer were used to count smaller and larger particles, respectively.

## 2.3 Merging Process

Merging procedures have usually been reported for merging SMPS and APS (Aerosol Particle Sizer) data, but here GRIMM data is merged with SMPS data. For a complete particle size distribution, paired hourly averaged particle number size distributions collected from the SMPS and GRIMM were merged. The merging procedure is based on the principle of converting the diameters of the GRIMM-derived data to a diameter matching the SMPS-derived data, in the region where the size distribution measurements overlap. The GRIMM measures the optical diameter $d_b^t$ whereas the SMPS measures the mobility diameter $d_a^t$ of the particles. Comprehensive descriptions of the procedure and



mathematics are given by DeCarlo et al. (2004) and Schmid et al. (2007). The GRIMM NSD are
translated onto the extended electical mobility diameter axis of the SMPS using equation (R1)
(Beddows et al. 2010; Liu et al., 2016; Ondracek et al., 2009).

$d_b^t = \dfrac{d_a^t}{X} \sqrt{\dfrac{C(d_a^t)}{C(d_b^t)}}$                      (R1)

The Cunningham slip correction factor is given by $C$ and the unknown variables such as the shape
factor of the particles are accounted for by a free parameter $X$ (given by equation R2) which is adjusted
until the tails of the SMPS and GRIMM NSD overlap each other giving a continuous NSD across the
particle size bins measured by the two instruments.

$X = \sqrt{\dfrac{\rho_e^t}{\rho_o}}$                          (R2)

The estimated transition-regime effective density $\rho_e^t$ (normalised by the unit density, $\rho_o$) typically
ranges from 0.77 to 2.56 g/cm$^3$ when aerodynamic diameter is used in merging. Detailed
information upon the effective particle density based on the geographical regions is seen in the Wu
and Boor (2021) study.

The merging algorithm (originally programmed in CRAN R) was implemented using Excel
spreadsheets and the solver tool minimised the separation between the tails of the overlapping SMPS
and GRIMM.  Due to the imperfect nature of the data, each of the merges was allocated a factor
indicating quality based on whether: (i) there is a successful fit; (ii) the scatter of the data across the
overlapping tails; (iii) the fraction of points on the tail falling onto the fitted curve; and (iv) how
smooth the overlap is (Table S1). The size bins overlap (300-700 nm) between Grimm and SMPS.
This process was repeated for the winter, summer and autumn data sets and any results failing the test





were either repeated or the data removed from the analysis. In all, only 8 samples from 1117 failed
to give an acceptable fit in the merge procedure.

**2.4      Data and Quality Management**
Data from SMPS and GRIMM were measured with 1-min resolution. In this study, data sets were
used by taking their hourly averages. Simultaneous measurement data from the SMPS and GRIMM
were used. The seasons were categorized as winter, autumn and summer.  The measurements were
taken in winter from 12 January 16:00 to 11 February 04:00, in autumn from 24October  16:00 to 11
November 10:00, in summer 16 May 19:00 to 05 June 15:00 in 2018. There were 709, 403 and 477
total pairs (hours) in the data sets in winter, autumn and summer, respectively. But 172, 43 and 257
pairs in winter, autumn and summer, respectively were excluded because of the non-availability of
data at that time. Data coverage is 76 % for winter, 95 % for autumn and 46 % for summer. Figure
S2 in the Supplementary shows hourly mean values of total particle counts for three seasons. In order
to evaluate day and night time PNC (particle number concentration) differences, the day and night
were defined as 07:00-19:00 and 19:00 – 07:00, respectively. All times  reported are local times
recorded in Indian Standard Time (IST; GMT+05:30).

R version 3.1.2 was used to analyse the data (R Core Team, 2015). Firstly, all data were checked for
clean-up of the robustness of the data sets, to detect anomalous records and take out the extreme
values. Data greater than the 99.5[th] percentile were deleted. Diwali time in 2018 (7th of November
2018 from 16:00 to 23:00) was taken out the date set in order to exclude its extreme effect on PSD
values. Particle number concentrations during Diwali time are given in the Supplementary, Figure
S3. There were some single gaps in the data matrixes. These missing data were replaced by linearly
interpolated values from the nearest bins to those samples.



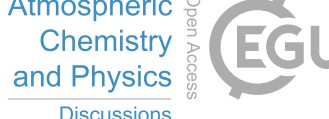

In the literature, PNCs measured below 1 µm are frequently split into three ranges: nucleation, Aitken
and accumulation (Gani et al., 2020). Size range of modes can be highly variable according to the
description of the nucleation size range and maximum measured size. Nucleation size ranges have
been described as below 30 nm (Masiol et al. 2016) or below 25 nm (Gani et al., 2020) or below 20
nm (Wu and Boor, 2021). Despite this, there are also limited studies on wide range PSD, some
evaluating wide range PSDs split into 4 ranges (nucleation, Aitken, accumulation and coarse) (Masiol
et al. 2016; Harrison et al., 2011). In this study, the modes have been aggregated into five size groups:
nucleation (15-20 nm), Aitken (20 -100 nm), accumulation (100 nm – 1 µm), large fine (1 µm – 2.5
µm) and coarse (2.5 µm – 10 µm) based on merged data using SMPS and GRIMM observations.
Ultrafine particles (UFP) are considered to be total PN counts of Nucleation and Aitken modes (<100
nm).

The particle mass was calculated for the SMPS+Grimm merged data, assuming a density of 1.6 g cm$^{-3}$
(Gani et al., 2020).  Figure S4 shows the comparison of PM$_{2.5}$ measured by SMPS+GRIMM and
TEOM-FDMS in Delhi for the three seasons. Figure S5 shows the comparison of PM$_{2.5}$ with relative
humidity measured by SMPS+GRIMM and TEOM in Delhi for the three seasons.  A good correlation
of the estimated particle mass with independent measurements with a co-located TEOM-FDMS was
observed, except in summer.

**3.      RESULTS**
**3.1      Particle Number and Size**
Table S2 gives the descriptive statistics of particle number counts (#/cm$^3$) calculated using every 1-
hour measurements for the nucleation, Aitken, accumulation, large fine and coarse modes between
15 nm and 10 µm in all seasons. Time series of total particle number counts are presented in Figure
S2. The average total PN levels were 36,730 #/cm$^3$ in winter, 29,355 #/cm$^3$ in autumn and 18,906
#/cm$^3$ in summer. Generally, the wintertime PN levels were higher than the other seasons.  The





wintertime PN levels of nucleation, Aitken and accumulation modes were ~1.5, 1.8 and 2.2 times
higher than in summer, respectively. Similar ratios were obtained by Guttikunda and Gurjar (2012)
in Delhi for particulate matter concentrations. This is attributed to the unfavorable dispersion
conditions, including low wind speed and low mixing height during the winter season. The autumn
PN levels of nucleation, Aitken and accumulation modes were ~1.5, 1.3 and 1.9 times higher than in
summer, respectively. The wintertime and autumn average PN levels are similar except for the Aitken
mode for which winter is 1.4 times higher than in autumn. However, for the large fine and coarse
modes the PN level was not markedly different between winter, autumn, and summer. Gani et al.
(2020) reported that the average PN levels were 52,500 #/cm$^3$ in winter, 43,400 in summer, and
38,000 #/cm$^3$ in autumn in Delhi measured in 2017. The differences in the magnitude of number
counts between the two studies are potentially explained by the difference in the sampling time and
changes in emissions.

Figure 1 shows a comparison of average particle number and volume and the contribution to total
PN. The average PV (particle volume) levels indicate that PV of the Aitken mode is highest in winter,
while the accumulation mode is highest in autumn and the coarse mode is highest in summer. The
contribution of UFP(<100nm) to numbers is highest in summer (57 %)  but their contribution to
volume is the lowest in autumn and summer ( <1 %). The contribution to both number and volume
of the accumulation mode is highest in autumn with 51 % and 75 %, respectively. UFP contributions
to total PV are below 1 % in Delhi. Furthermore, it can be seen clearly that the coarse fraction of
particles dominates in summer, while the accumulation mode dominates in autumn and winter.
Wu and Boor (2021) analysed the PSD observations made between 1998 and 2017 in 114 cities in 43
countries around the globe. They reported that there are significant variations in the magnitude of
urban aerosol PSD among different geographical regions. The main finding of their study is that the
number PSD in Europe, North America, Australia, and New Zealand are dominated by nucleation-
and Aitken-mode particles while in Central, South, Southeast and East Asia they are dominated by

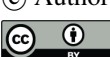



the substantial contribution from the accumulation mode, which is consistent with our finding.  Pant
et al. (2016) report mass size distributions for particulate matter sampled by cascade impactor in Delhi
in winter.  The dominant modes appear at around 3-4 µm and 0.6 µm, with a lesser peak at 0.2 µm
aerodynamic diameter.  These are respectively in the coarse (former mode) and accumulation (latter
two modes) ranges as classified in the current study.  The largest component of mass was in the
accumulation mode, and the distribution fits well with the pattern of data seen in Figure 1.  Major
components of the coarse fraction were Al, Si, Ca and Fe (Pant et al., 2016), suggestive of soil and
street dust as major contributors.  The elements most notably in the accumulation fraction were Cu,
Zn, Pb and Sb, indicative of non-exhaust traffic emissions and metallurgical sources, and S, which
showed a major peak due to sulphate, peaking at 0.9 µm (Pant et al., 2016).

**3.2    Diurnal Change**
Figure 2 shows the diurnal variation of particle number concentrations and of  $PM_{2.5}$, BC, NO and
$NO_2$ for each season (excluding the day of Diwali), and the normalized time variations of all particle
fractions are given in Figure S6. Figure S7 represents the diurnal variation of meteorological
parameters. In general, there are large differences of PN levels between cold seasons (winter and
autumn) and warm season (summer) for  nucleation size particles. Coarse mode particle numbers in
the summer are higher than in winter and autumn, except in the evening time. For autumn and winter,
particle counts are similar from 7 am to 7 pm (daytime). However, from 7 pm to 7 am particle counts
in winter are higher than in autumn. The lowest levels for all modes were present during the afternoon
in all seasons (2-4 pm), followed by highest levels during the night in winter (after 8 pm).  The winter
and autumn diurnal profiles had two peaks for below 1 µm particle size in the morning and evening
corresponding to the traffic rush hours. But in the summer the same peaks for nucleation, Aitken and
accumulation modes are seen although of smaller magnitude, and one hour earlier comparing to the
winter and autumn.  Pant et al. (2016) reported the diurnal variation of traffic at one of the major
arterial roads in Delhi and Dhyani et al. (2019) reported on traffic-related emission. Figure S8 shows





the diurnal variation in traffic at a major road intersection in Delhi. Cars, two/three wheelers, bus and
LCV (light commercial vehicle) fleet numbers increase in the morning, persist throughout daytime
and start to decrease at 22:00. Due to the prohibition of access for heavy-duty diesel vehicles to central
Delhi from 6:00 am until 11:00 pm in the night, during the daytime including the traffic rush hours
the HCV (heavy commercial vehicles) number is at its lowest level (Figure S8 or Dhyani et al. 2019).
Small midday PN peaks were observed during the summer in the nucleation, Aitken and coarse
modes. Another study conducted in Delhi reported the same midday peaks in the warm season and
the highest levels in the cold season (Gani et al. 2020), which may be related to bus and LCV
emissions at midday.

Figure 3 shows the differences in diurnal variations of total PN levels between the weekday and
weekend. These are based upon a small dataset, and hence the rather small differences within a season
may not be meaningful. In winter the PN levels on Saturday and Sunday are higher than on the
weekdays during the night (from 8 pm to 10 am the next day). However, after the morning rush hour
peaks, during the daytime the PN levels are the same for all days. The diurnal variation of PN in the
autumn shows no significant differences among the days with the same main peaks in the morning
for all days, although highest on Saturday. There is a flattened peak (from 8 to 10 am) in the morning
rush hour for the weekday while there are pointed peaks at approximately 9 am on Saturday and
Sunday in winter and autumn. Measurements made during the summer period are very limited. Due
to there being only 4 full days and 9 half days of measurements, it is very hard to draw any
conclusions. Even so, there are indications of a weekday traffic effect upon the PN levels in summer.
There is only one day of measurements on a Sunday (3rd June 2018) and it shows the midday peaks.
Overall, despite seasonal differences, there appears to be a strong influence of light duty road vehicles
upon the diurnal profiles, reflecting traffic volumes, with an impact of heavy duty vehicles upon
nighttime concentrations of all particle fractions.



NPF events present variable seasonality for different areas, though in most cases they appear to be
more frequent during spring or summer (Salvador et al., 2021). Gani et al. 2020 studied long term
PSD in Delhi and have stated that they did not see any NPF during the winter or autumn seasons in
Delhi. In this study, the identification of NPF events was conducted manually using the criteria set
by Dal Maso et al. (2005) and used by Bousiotis et al (2019; 2021). The data were analysed visually
on a day-to-day basis: each 24-hour period, from midnight to midnight. According to these criteria, a
NPF event is considered when: a distinctly new mode of particles appears in the nucleation mode size
range, prevails for some hours, and shows signs of growth. These are the initial criteria used in
identifying the events. Following that, as the dataset starts from a rather large size (15 nm), to be
more confident about the events and not to confuse them with pollution events, high time resolution
data for $NO_x$ as well as the fluctuations of the condensation sink were also used to identify pollution
events affecting particle concentrations which were not considered. Hence, while we checked the
particle size distributions for the NPF events, we also looked at the levels of pollutants to ensure that
what was attributed to a NPF event was not particles from pollution / direct emissions. By considering
the pollution levels and condensation sink we can reduce the possibility of including particle
formation events that are not associated with secondary formation. After analysing all data,
measurements from only one day during the measurement campaign were compliant with the criteria
set as a class Ia NPF event. Figure 4 presents the contour plots of average diurnal variation for all
seasons and for the NPF event on 3rd June. NPF may be suppressed due to very high pre-existing
aerosol concentrations (Kanawade et al., 2020; Gani et al. 2020) during severe air pollution episodes
in Delhi. This suppression effect has also been observed in European cities (Bousiotis et al., 2019;

346    2021).


**3.3      Day and Night Time Differences in PN and PV**
Table S3 presents the summary statistics of the particle number and mass levels derived from merged
particle number data and BC, $NO_x$ and $PM_{2.5}$ at night and day for each season, excluding Diwali.
Figure 5 shows the particle number comparison of all modes at night and day seasonally. In both



night and day, the nucleation counts are approximately the same in autumn and summer (N/D=1.1
and 1.0), and a little higher at night in winter (N/D=1.3). But in the night, Aitken and accumulation
counts are higher than in the day by factors of 1.4 and 1.5 times in summer, 1.2 and 1.5 times in
autumn, respectively and approximately 2 in winter. While the coarse mode PN counts are
approximately the same for all seasons and day / night, the large fine PN level in the nighttime are
significantly higher (1.7) than in the daytime in summer. It seems that in the nighttime high PM
concentrations are due to the increasing Aitken and accumulation modes occurring from coagulation
of nucleation mode particles, condensation of low volatility species or hygroscopic growth. In
addition, biomass burning and older diesel vehicles can contribute significantly to particles in these
fractions (Kumar et al., 2013; Chen et al., 2017; Gani et al., 2020). Meteorological factors can also
profoundly affect the PN levels in daytime and nighttime. The differences of wind speed between day
and night in summer are lower than in winter and autumn (Figure S7). Higher wind speed, and lower
humidity, may favour the resuspension of coarse dust as a dominant mechanism in the summer.
Seasonal changes in mixing depths are surprisingly small (Figure S7) and hence unlikely to have a
major influence.

Overall, for the daytime for all seasons, hourly averaged UFP (<100nm) concentrations are usually
less than the nighttime, however the UFP contribution to the $PN_1$ (55 % in day, 50 % in night for
winter; 52 % in day, 45 % in night for autumn; 58 % in day, 56 % in night for summer) and $PN_{10}$ (38
% in day, 38 % in night for winter; 40 % in day, 33 % in night for autumn; 36 % in day, 33 % in night
for summer) are mostly slightly higher in the daytime. Similarly, Gani et al. (2020) have reported the
highest contribution (of UFP to PNC) in the daytime compared with the nighttime in Delhi. Due to
the difference of PN size range (they measured down to 12nm), they found the UFP contribution to
PNC higher than in the present study.





### 3.4 Size Distributions

Figure 5 shows the average particle number size distributions in three seasons in Delhi. Volume and Area distributions are shown in the Supplementary Materials in Figure S9. The highest number concentrations are seen in winter, followed by autumn, and then summer. Although the number concentrations of particles below 200 nm are far greater in winter those between 200 and 600nm are greater in autumn, within the accumulation mode. The winter and summer PSD show modes at approximately 100 nm but the autumn PSD shows the mode at approximately 200 nm. This could be due to changing sources of particles in Delhi between seasons (Jain et al. 2020), in addition to (differing) aerosol dynamical processes. The Delhi atmosphere is more polluted comparing with most other cities based on particle number and mass (Harrison, 2020). This will cause a tendency for particles to grow more rapidly by coagulation and condensation (Harrison et al., 2018), but this might be expected to occur in all seasons.

As described above, in Delhi the nighttime particle concentrations are markedly higher than the daytime concentrations. The PSD changes for each hour of the day across all three seasons were analysed (Figure S10) and categorized. Figure 7 presents the PSD differences between daytime and nighttime and shows the variation in PSDs within the day in all seasons. The main difference between day and night in winter is only the number concentration, with little change in the mode size between day and night, while the PSDs in summer and autumn show bimodal distributions with modes at approximately 30 and 140 nm in summer, and 35 and 200 nm in autumn. When we focus on PSD during the daytime, it can be clearly seen that the modes are manifest at different times: In winter, while the PSD shows the same mode at approximately 100 nm from 8 am to 2 pm, the mode in the afternoon (from 2 pm to 6 pm) drops slightly in size (70 nm). In the morning and afternoon there are two small peaks at 60 nm and 40 nm for the Aitken fraction and 170 and 130 nm for the accumulation fraction in autumn. During the day in summer, there are two peaks at approximately 30 nm from 10 am to 6 pm. This may be associated with summer nucleation events and NPF on 3rd June 2018 (Figure



4). Furthermore it may be related to the growth of particles from 10 am to 2 pm in autumn and
summer. The full reasons for these changing PSDs are not clear, and it would be unwise to attempt a
detailed interpretationof a very small dataset.

Figure 8 shows the average geometric mean diameter (GMD) change with hour of the day. Two
overall periods of GMD increase are observed. One of them is in nighttime in all seasons with GMD
growing at between 4.6 nm/hour in summer and 6.2 nm/hour in winter. The particle growth in autumn
is predominantly (when compared to the winter and summer) both late in the night (from 0 am to 5
am) and in the morning (from 8 am to 12 pm). Considering the PSD trend in autumn (Figure 7), the
GMD rises at 9 nm/hour from morning to noon. Similar results were obtained in the USA (Kuang et
al., 2012), Canada (Jeong et al., 2010; Iida et al. 2008), Italy (Hamed et al., 2007) and Japan (Han et
al. 2013). However the calculated GMD growth rate is smaller than that calculated by Sarangi et al.,
(2015; 2018) in Delhi, by Kalafut-Pettibone et al. (2011) in Mexico City and by Zhang et al. (2011)
in Beijing. The changing GMD with time in Delhi could be the result of changing sources, and/or of
dynamics. Nocturnal growth may be the result of reducing temperatures and increasing RH causing
vapour condensation (Sarangi et al., 2018). Morning growth may be due to oxidation processes
leading to production of less volatile vapours which then condense onto the particles (Sarangi et al.

422 2018).


Figure 9 gives the average particle number, volume, area and mass size distribution for all seasons.
While the number size distributions have one mode, two peaks are observed in volume distributions,
centered at 0.5 µm and 6 µm. These relate to two different main sources, which might be secondary
aerosol (such as sulphate at high RH) in the fine mode and road dust resuspension, soil or construction
dust for the coarse mode (Pant et al. 2016). In winter and autumn fine mode particle volumes are
higher than the coarse mode. However, in summer the coarse mode particle volumes are higher than
the fine particle level. In a recent paper, Thamban et al. (2021) show that modes in the mass size



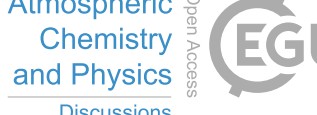

distributions of HOA, SVOOA, BBOA and LVOOA measured by aerosol mass spectrometry are
typically in the range 300-600nm vacuum aerodynamic diameter, very consistent with the peaks seen
in the mass distributions in Figure 9.

Hama et al. (2020) obtained the spatiotemporal characteristics of daily-averaged air pollutants and
concluded that the particulate matter mass ($PM_{10}$ and $PM_{2.5}$) is dominated by local sources across
Delhi. The main local air pollutant sources in Delhi include traffic, construction, resuspension of dust,
diesel generators, power plants, industries and biomass burning (Kumar et al., 2013; Nagpure et al.,
2015; Hama et al., 2020).

All average PSD graphs show an increasing trend in PNC at particle sizes below 19 nm particle
diameter. SMPS measurements in this study were conducted only above 15 nm. So, the peak particle
size within this size range cannot be seen. However, the clear increase in particle number below 19
nm indicates that another source may be important in Delhi. This small mode and bimodal PSD during
the day (Figure 7) may be associated with the road transport vehicle types in Delhi.  Despite the diesel
restriction during the rush hours and conversion of the public transport vehicles to CNG, several
studies have reported that $PM_{2.5}$ concentrations have been remaining steady or are slowly increasing
in India, especially in the winter and autumn seasons (Babu et al., 2013; Balakrishnan et al., 2019;
Dandona et al. 2017; Kumar et al., 2017).

The fuels used in Delhi's traffic fleet are petrol, diesel, CNG and LPG. Legislation limits the sulphur
content of the fuel to 50 ppm in diesel as per Bharat Stage IV. The diesel vehicles are not required to
be fitted with particle traps. The technology of the gasoline vehicle fleet varies as vehicle engine
capacity changes. Cars, two/three wheelers, bus and LCV fleet volumes are high during the day. Due
to the time restrictions on trucks/heavy good vehicles entering the city, during the daytime the HCV
number is at its lowest level (Figure S8).





Previous published studies indicate that emissions of particles from CNG vehicles (Euro 4, 5, 6) with
diameter greater than 23 nm are as low as a diesel particle filter equipped vehicle, and an order of
magnitude lower than gasoline vehicles (Kontses et al. 2020; Giechaskiel, et al. 2019; Magara-Gomez
et al. 2014; Schreiber et al. 2007), and CNG vehicles mainly emit nuclei-mode particles  (Zhu et a.,
2014; Toumasatos et al. (2020).   Zhu et al. (2014) calculated size-resolved particle emission factors
from on-road diesel buses and CNG buses and reported that the PSD of diesel buses dominate the
accumulation mode diameters of 74-87 nm while the PSD of CNG buses dominated the nucleation
mode with modes at 21-24 nm. Total PN emissions of diesel buses per vehicle were 4 times higher
than the level of CNG buses. However, the PN level in the nucleation mode (15-25 nm) of CNG buses
was 1.7 times higher than from the diesel buses in the nucleation mode. Toumasatos et al. (2020)
studied the particle emission performance of the Euro 6 CNG and gasoline vehicles and discussed the
current EU cut-off solid PN size threshold of 23 nm. The results revealed that PN>23 nm represented
43 % of PN>10 nm and 8 % of PN>2.5 nm for gasoline vehicles and 7 % of PN>10 nm and 1 % of
PN>2.5 nm for CNG vehicles respectively. These studies of emission PSDs show that a significant
number of particles reside below the EU lower measurement limit of of 23 nm, and many are even
smaller than 10 nm.  These probably contribute to the mode seen just appearing at the extreme small
particle limit of Figure 9.

When the PSD results measured in Delhi are compared with the main emission categories in the
literature (Kumar et al., 2013; Vu et al., 2015), it seems that the average size distributions measured
in the atmosphere in Delhi are much coarser, which is presumably due to condensation and
coagulation, or it could be that secondary particles dominate over the primary emissions. Pant et al.
(2016) hypothesised that the main accumulation mode peak in their winter measurements arose from
aqueous droplet evaporation, although this mechanism would be unlikely to explain the mode seen
in the summer data.  Thamban et al. (2021) have also reported particle growth in the Delhi atmosphere
from condensation of organic compounds formed from oxidation processes.





Previous studies have attempted to quantify the relative contribution of primary and secondary
sources to the total and mode-segregated particle number concentrations (Kulmala et al. 2021;
Casquero-Vera et al. 2021; Hama et al. 2017; Kulmala et al. 2016; Rodríguez, & Cuevas, 2007).
Rodríguez and Cuevas (2007) first presented the methodology for the separation of traffic related
primary aerosol particles from the total using the BC as the main tracer of traffic. The method was
tested in this study, but did not prove appropriate as the BC sources in Delhi are more complex, and
arise not only from traffic. The BC diurnal trend (Figure 2) does not show the rush hour peaks, and
reflects mostly the combustion activity at night, presumably including the heavy duty diesel
emissions. A recent study by Kulmala et al. (2021) used $NO_x$ as a tracer of primary sources. Figure 2
shows that only the $NO_2$ diurnal trend in autumn is clearly related to traffic sources. Furthermore the
sources of BC and $NO_x$ are largely the same, as judged from the high similarity between BC and $NO_x$
diurnal trends (Figure 2).

**3.5      Correlations of PN with $NO_2$, NO, and BC**
Figure 10 shows the correlation coefficients between the hourly average PNs of five particle size
fractions and NO, $NO_2$, and  BC measured in Delhi. Nucleation mode PN is better correlated with the
Aitken mode PN in winter and summer despite the lower correlation in autumn. The correlations
among >1 µm size fractions are higher in summer than winter and autumn. Tyagi et al. (2016) stated
that the major source of $NO_x$ emissions is vehicle exhaust and power plants in Delhi. Furthermore,
studies have reported that approximately 80-90 % of $NO_x$ and CO are produced from the transport
sector in Delhi (Gurjar  et al., 2004; Gulia et al., 2015; Tyagi et al., 2016; Hama et al., 2020). As seen
in Figure 2, the $NO_2$ diurnal trend is very similar to nucleation and Aitken particle trends, especially
in the autumn. $NO_2$ peaks in autumn in the traffic rush hours are larger than in winter and summer.
In addition, there are no significant correlations between $NO_2$ and NO or BC in autumn (0.02 for NO,
0.03 for BC) compared to the summer (0.73 for NO, 0.61 for BC) and winter (0.37 for NO and 0.28
for BC) (Figure S11).  NO and BC diurnal trends show the same higher level in the night (Figure 2)





and also, they have higher correlation coefficients (0.78 in winter, 0.77 in summer, 0.72 in autumn)
for all seasons, similar to the accumulation mode particle counts (Figure 2, Figure 10, Table S2). $NO_2$
and <100 nm particles may be associated with traffic sources, while the NO and BC and <1 µm
particles could be associated with biomass burning, industry, (small generator) power generation, or
possibly also with diesel vehicles.

**3.6     Wind Effects**
Figure S12 represents polar plots of BC, NO and $NO_2$ measured in Delhi. This shows no consistent
pattern. There are differences between the pollutants in terms of directional and wind speed
associations, and for each pollutant / season.  There is no obvious indication of a strong local source
influence, typically manifest as an intense area in the very centre of the plot circle. The plots for the
particle size fractions (Figure 11) also show little consistency between seasons for a given size
fraction.  Within a season, however, adjacent size fractions often show a similarity of behaviour
(consistent with their correlations, see above) but this similarity does not extend across all size ranges
within a season.

**4.     CONCLUSIONS**
This study serves to highlight the remarkable complexity of airborne particulate matter in Delhi.  The
size distributions show marked seasonal changes, with coarse particles dominant in summer, but not
in the cooler seasons, when the accumulation mode dominates.  The measured size distributions show
a fine mode aerosol with a considerably larger modal diameter than that typically seen in western
countries, and larger than the modal emission size from major source categories.  It appears the that
the high particle concentrations and chemically reactive atmosphere are promoting rapid coagulation
and condensational growth of particles, and therefore the measured size distributions are driven more
by aerosol dynamical processes than source characteristics.  Growth via a liquid droplet phase in the
cooler months may also occur.  There is little evidence for a contribution of new particle formation





(although the summer season dataset is small), consistent with earlier work by Gani et al. (2020).
Another notable feature is the apparent complexity and seasonal variability of sources of NO, NO$_2$
and BC, pollutants which can often be used to identify or locate sources of emissions. This is reflected
in the various particle fractions, which generally correlate poorly with the other pollutants and with
other than proximate size fractions.

The diurnal variation of all particle fractions is strongly suggestive of a road traffic influence,
especially in the winter campaign. This appears strongly influenced by the emissions of heavy duty
diesel traffic which is only able to access central Delhi at night. A size mode of <15 nm may well
be attributable to vehicles using LPG/CNG fuels. However, the seasonal variability of the geographic
distribution and wind speed dependence of sources revealed by the polar plots is strongly indicative
of many other sources also contributing to all size fractions of particles.

**DATA ACCESSIBILITY**
Data supporting this publication are openly available from the UBIRA eData repository at
https://doi.org/10.25500/edata.bham.00000730

**AUTHOR CONTRIBUTIONS**
This study was conceived by RMH. WJB managed the research programme, and MSA and LRC
collected the data. DCB and UAS led the data analysis with contributions from JB and DB. UAS
and RMH co-authored the first draft. ZS and all co-authors provided comments and revisions.

**COMPETING INTERESTS**
The authors declare that they have no conflict of interest.






**FINANCIAL SUPPORT**
This research has been supported by the Natural Environment Research Council NE/P016499/1.

**ACKNOWLEDGEMENTS**
We are thankful the Scientific and Technical Research Council of Turkey (TUBITAK) (grant
number 1059B191801445) to support the author Ulku Alver Sahin to work on the ASAP project.
We would also like to acknowledgement the IIT Delhi team under the Principal Investigator-ship of
Professor Mukesh Khare who provided all the facilities including space and logistical help during
the experiment.





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





**FIGURE CAPTIONS:**

**Figure 1.**   Comparison of average particle number counts (#/cm$^3$) for nucleation, aitken, accumulation, large fine and coarse modes of PM between 15 nm and 10 µm in all seasons, and the volume contributions for comparison.

**Figure 2.**   Average diurnal variation of particle number counts for nucleation, Aitken, accumulation, large fine and coarse modes and PM$_{2.5}$, BC, NO and NO$_2$ in autumn, summer and winter.

**Figure 3.**   Average diurnal variation of total particle number counts (between 15 and 10 µm) for weekday average (Monday to Friday), Saturday and Sunday in Delhi. (Summer data are very limited. There are no data on Friday afternoon, night and Saturday early morning (Figure S6)).

**Figure 4.**   Diurnal contour plots for particles derived by SMPS between 15 and 660 nm averaged for each season (a: winter, b: Autumn and c: Summer) and for 3 June 2018 data when a NPF event probably occurred (d), the solid line showing the NO$_x$ mixing ratio. Note the different scales for the seasons presented.

**Figure 5.**   The hourly average of day and night particle numbers for all modes from the wide range particle sizes derived from the merged data. UFP =Nucleation +Aitken, PN$_1$ = UFP+Accumulation, PN$_{10}$= PN$_1$+Large Fine+Coarse.

**Figure 6.**   Seasonal average (line) and standard deviation (shadow) of particle number size distributions.

**Figure 7.**   Hourly average day and night (left side) and during day hours (right side) particle number distributions in autumn, summer and winter in Delhi.

**Figure 8.**   Diurnal change of the geometric mean diameter (GMD) calculated for winter, autumn and summer seasons.  Growth rates (nm/hour) are calculated from dGMD/dt.

**Figure  9.**   Hourly average particle number, volume and area distributions in the winter (a), autumn (b) and summer (c) in Delhi.

**Figure 10.**   Correlation coefficient (R) between the hourly average PNs of five particle size fractions (left side) and NO, NO$_2$, BC (right side).

**Figure 11.**   Polar plots of PNs (#/cm$^3$) for five particle size fractions in winter, autumn and summer in Delhi.


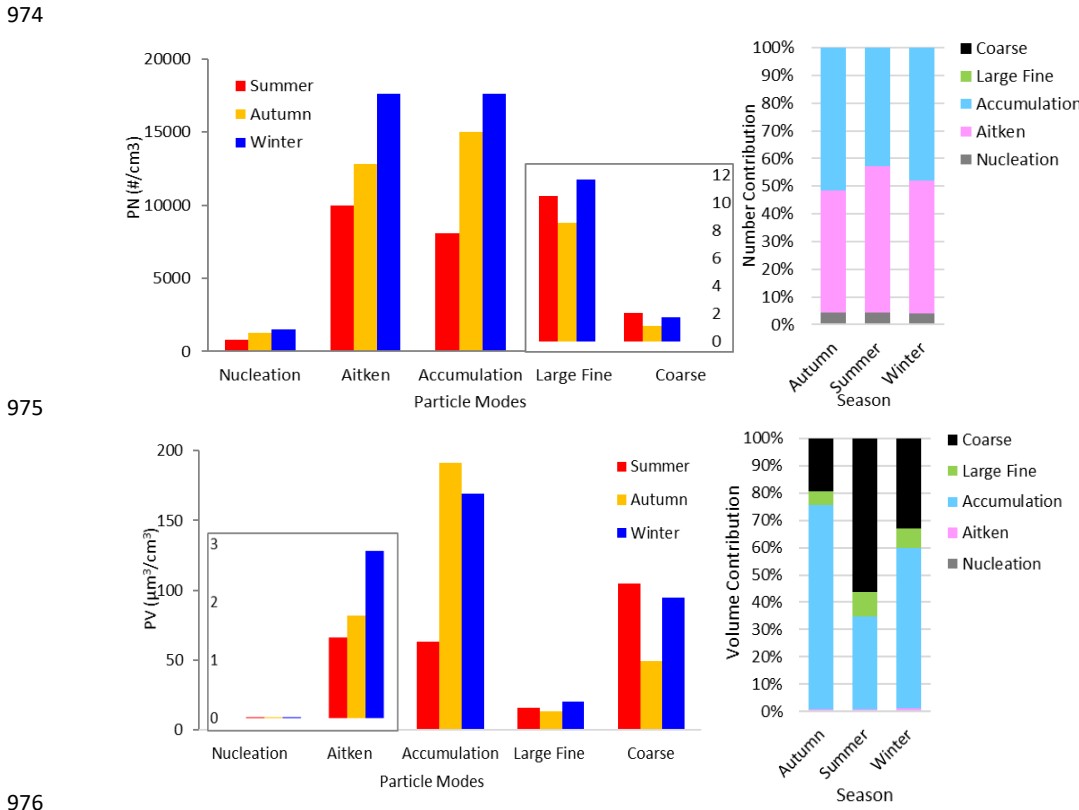




**Figure 1.** Comparison of average particle number counts (#/cm$^3$) for nucleation, aitken, accumulation, large fine and coarse modes of PM between 15 nm and 10 µm in all seasons, and the volume contributions for comparison.










**Figure 2.** Average diurnal variation of particle number counts for nucleation, Aitken, accumulation, large fine and coarse modes and PM$_{2.5}$, BC, NO and NO$_2$ in autumn, summer and winter.







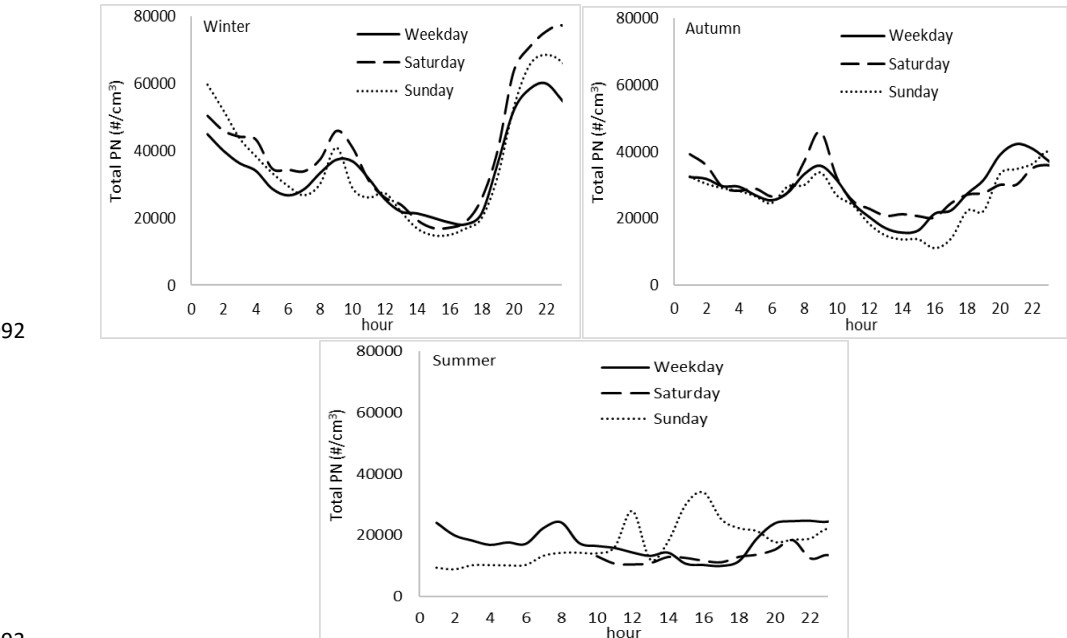



**Figure 3.** Average diurnal variation of total particle number counts (between 15 and 10 µm) for
weekday average (Monday to Friday), Saturday and Sunday in Delhi. (Summer data are very
limited. There are no data on Friday afternoon, night and Saturday early morning (Figure S6)).





a)   Winter                                   b) Autumn

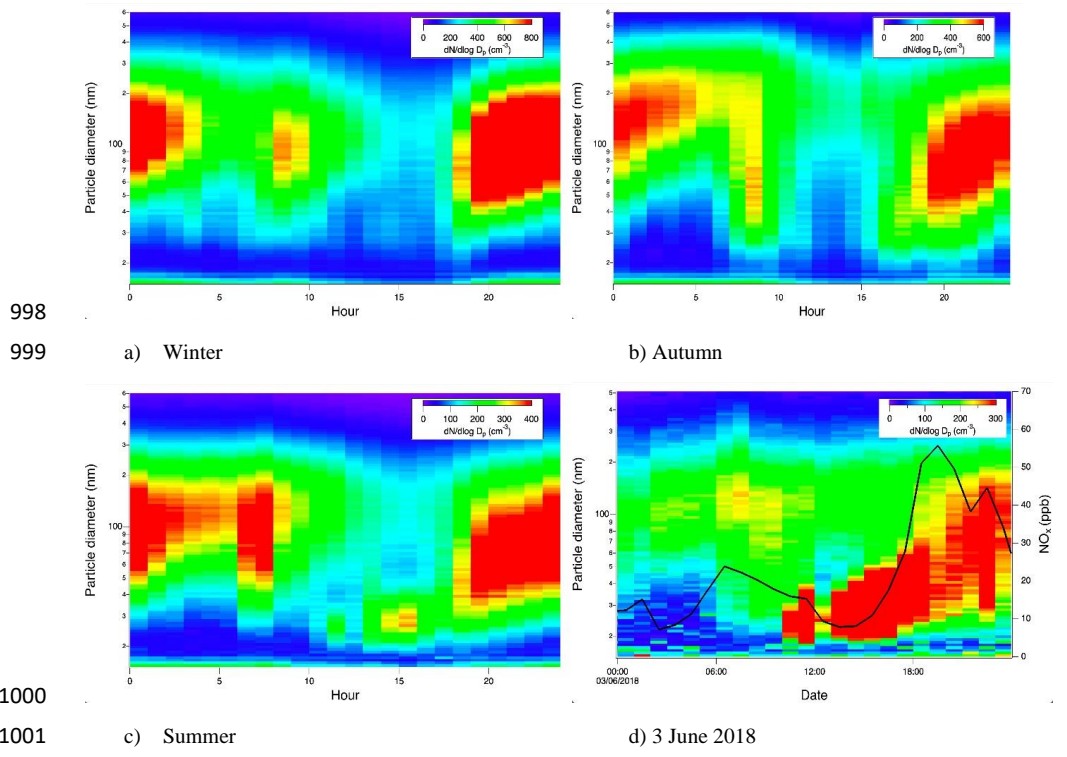


c)   Summer                                   d) 3 June 2018

**Figure 4.** Diurnal contour plots for particles derived by SMPS between 15 and 660 nm averaged for
each season (a: winter, b: Autumn and c: Summer) and for 3 June 2018 data when a NPF event
probably occurred (d), the solid line showing the $NO_x$ mixing ratio. Note the different scales for the
seasons presented.



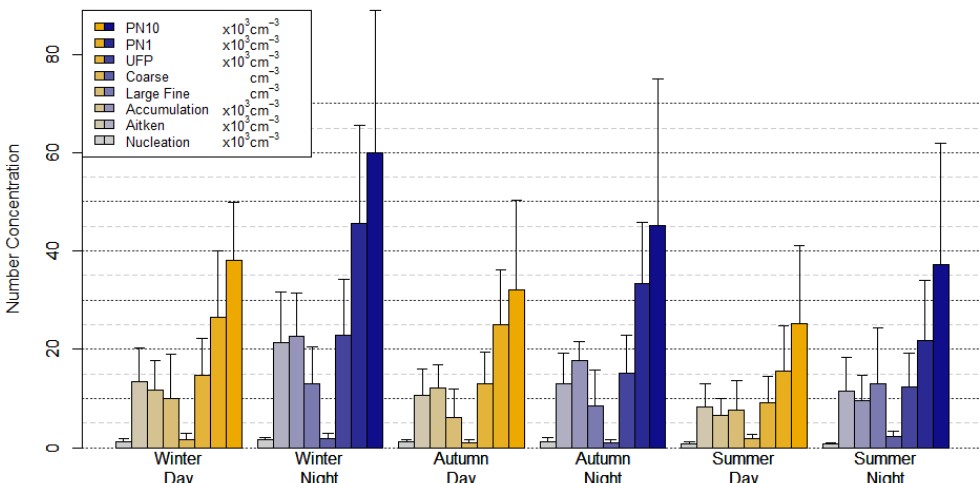

**Figure 5.** The hourly average of day and night particle numbers for all modes from the wide range particle sizes derived from the merged data. UFP =Nucleation +Aitken, $PN_1$ = UFP+Accumulation, $PN_{10}$= $PN_1$+Large Fine+Coarse.





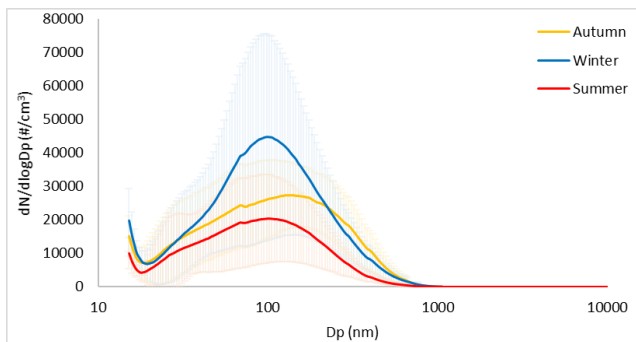


**Figure 6.** Seasonal average (line) and standard deviation (shadow) of particle number size
distributions.




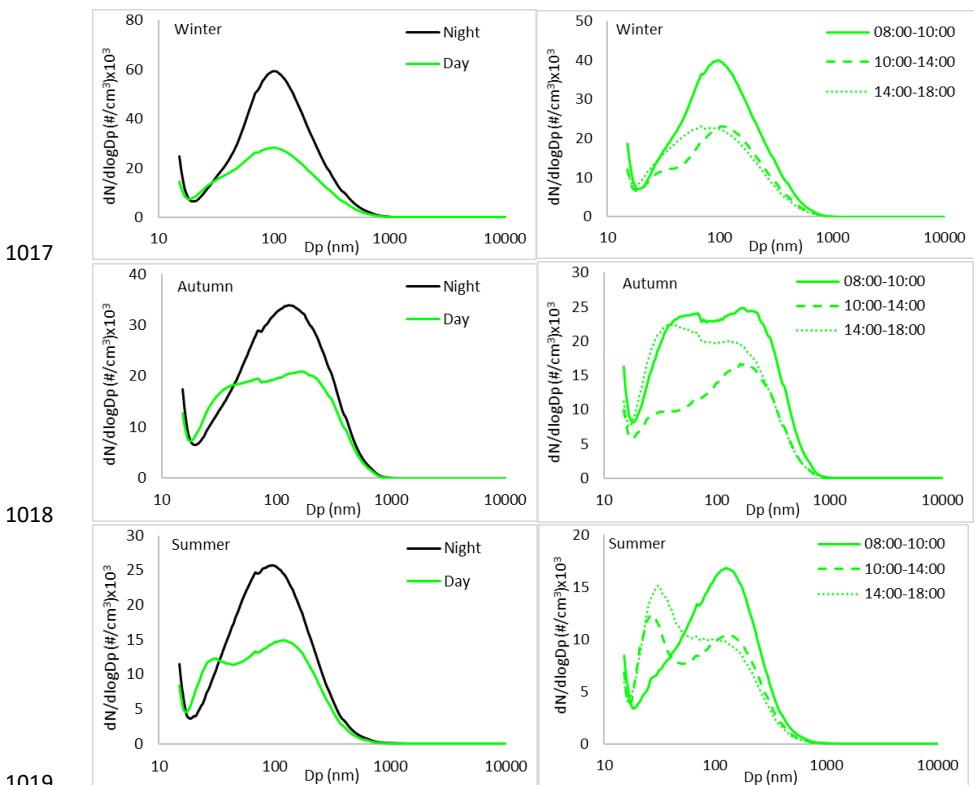



**Figure 7.** Hourly average day and night (left side) and during day hours (right side) particle number
distributions in autumn, summer and winter in Delhi.





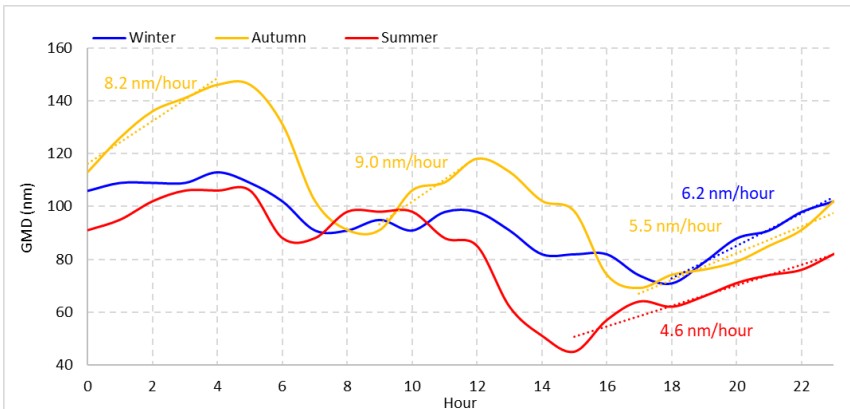


**Figure 8.** Diurnal change of the geometric mean diameter (GMD) calculated for winter, autumn
and summer seasons. Growth rates (nm/hour) are calculated from dGMD/dt.






**Figure 9.** Hourly average particle number, volume and area distributions in the winter (a), autumn (b) and summer (c) in Delhi.










**Figure 10.** Correlation coefficient (R) between the hourly average PNs of five particle size fractions (left side) and NO, $NO_2$, BC (right side).








Winter  Autumn  Summer



**Figure 11.** Polar plots of PNs (#/cm$^3$) for five particle size fractions in winter, autumn and summer in Delhi.