# Peer review of "Measurement Report: Interpretation of Wide Range Particulate Matter Size Distributions in Delhi Ülkü Alver Şahin1, Roy M Harrison2,a, Mohammed S. Alam2,b David C.S. Beddows2,3, Dimitrios Bousiotis2, Zongbo Shi2 Leigh R. Crilley<su"

_Atmospheric Chemistry and Physics, 2021_

## Referee Comment (RC2)

The MS as a measurement report mainly deals with deriving wide-range particle number size distributions from a mobility particle size spectrometer and a particle optical counter, and also presents its application, results and conclusions for Delhi. Its basic ideas, topic and outcomes are timely, important and of interest for the research community. Its evaluation methods are mostly plausible. At the same time, the MS contains several conceptional errors, which are severe, and are misleading in many ways. They cannot be tolerated in a journal with an impact factor of 6.546 and should be definitely corrected or removed before any reliable evaluation of the MS could be finalised. There is also a large number of smaller discrepancies in the text and figures which should be improved and handled. The list below gives only examples of them.

Major comments

1. The authors may want to emphasize better the need for and advantages of wide-range particle number size distributions.
2. The abbreviation PM expresses particulate matter, thus the set of aerosol particles (and not their any property). It is used over the whole MS in a fuzzy manner, sometimes hinting at the PM mass. A very clear distinction between the PM mass and particle numbers should be made throughout the text since the main objectives of the MS are related to particle number concentrations and size distributions. This seemingly small discrepancy results in several misleading sentences and erroneous formulations. Examples are L90-92 and L95-96: with regard to the mass or particle number? With respect to this, $PM_{10}$ volume is correct, while the formulation "$PN_{10}$ number" is unusual, not consequent and, therefore, confusing (L50).
3. The situation is somewhat similar with the particle size distribution (L98). The expression is meaningless in its present form. Size distributions are exclusively related to properties of particles, which are missing from the expression. There are mass size distributions, surface area size distributions, particle number size distributions, etc. This should be added at many places in the MS and the title should also be changed accordingly.
4. The size distributions consist of modes (peaks which extend from $-\infty$ to $+\infty$). Their significant content or area can be approximated by a size range. The modes are: nucleation, Aitken, accumulation and coarse, while the classical size fractions are ultrafine, fine and coarse or $PM_{10}$, $PM_{2.5}$, $PM_{1.0}$ and etc. The size fractions can be defined freely. The authors should not, however, mix the modes and the size fractions. Accumulation size fraction could be preferred to accumulation mode in this aspect. In addition, in L54-L55: which mode?; in L380: area distribution?

5. The authors are requested to discuss the dependency of the particle density on the size (L232), to describe how the particle volume was actually calculated from particle number and particularly, what the resulting uncertainties from the conversion were (L380), and specify the method of deriving GMDs (L409+).

6. The diurnal variation of BC (Fig. 2) and the increasing part of the particle number size distributions with decreasing particle diameter in the range <ca. 20 nm (Fig. 6) could also be explained and discussed (better).

Minor comments

7. Abbreviations TEOM-FDMS (L136-137), HOA, SVOOA, BBOA and LVOOA (L431) are not resolved, UFP is defined several times (L229, L262, L368), SMPS is explained not at its first appearance in the body text (L146 vs. 158).

8. The size distribution and other properties obtained by merging the SMPS and Grimm optical spectrometer data are better to be called SMPS-OP data instead of SMPS-Grimm data, with OP standing for particle optical counter (L164, L178, L199+).

9. Part L201-L206 belongs more to the description of the experimental methods than to section Data and Quality Management. Part L220-L230 contains extensive repetitions, and its rest should be shifted to methodology.

10. The authors should revisit their rounding off strategy. Example: instead of 36730 cm$^{-3}$ give 3.7×10$^3$ cm$^{-3}$ (L244).

---

## Author Response (AR1)

**MS No.:** acp-2021-860 **MS type:** Measurement report
**Title:** Measurement Report: Interpretation of Wide Range Particulate Matter Size Distributions in Delhi
**Author(s):** Ülkü Alver Şahin et al.

**RESPONSE TO REVIEWERS**

**REVIEWER #1**

This manuscript presents an analysis of the particle number concentrations in the range of 15nm to 10μm during winter, autumn, and summer of 2018 in Delhi. The particle size distribution measurements presented here are based on a scanning mobility particle sizer (SMPS) and a GRIMM spectrometer/dust monitor which were deployed 15m above ground and 100-200m away from a major arterial road in Delhi. Consistent with past studies, the mode diameters observed in Delhi during polluted periods (e.g., winter/autumn early-morning/nighttime) are some of the highest observed anywhere on the planet. Overall, this measurement report article is well written and generally easy to follow. However, I do think that some revisions are required to the manuscript.

Please consider the following comments which may improve the manuscript:
Major comments:
1. Mixing layer height (MLH) is an important factor in both seasonal and diurnal variation of pollutant concentrations in Delhi (Gani et al., 2019). The interpretation of Figure S7 "Seasonal changes in mixing depths are surprisingly small (Figure S7) and hence unlikely to have a major influence." (page 14, line 365) is not convincing. Please consider the following:

a) Figure S7 does show that autumn/winter has longer periods with shallow inversions.
**RESPONSE:** We are pleased to acknowledge this and have added a sentence accordingly in L404. [Line numbers refer to the tracked manuscript.]
"Autumn and winter also have longer periods with low mixing heights, also seen in Figure S7."

b) The major increases/decreases in the diurnal plots of pollutants (Figure 2) seem to be consistent with decrease/increase in MLH diurnal plots for each season.
**RESPONSE:** We agree with this interpretation, which is demonstrated by Fig R1, below. We have added the sentence "However, the major increases and decreases in the diurnal plots of pollutants (Figure 2) are consistent with the diurnal plots of MLH (Figure S7)" in L402 in the MS.

[Figure]

Figure R1: The relation of diurnal variation for MH and pollutants.

c) Please include data source and averaging method (arithmetic mean, median?) for MLH data. You can also consider the implications of the averaging method used to the interpretations of the influence of the MLH to pollutant concentrations.

**RESPONSE:** The diurnal variation figures (which use the arithmetic mean values) of all meteorological parameters were created using the data measured at the same time as the PN measurements in this study for all seasons. The MLH value has been mostly calculated using the arithmetic mean in the literature, such as in Gani et al. (2020). We compared the arithmetic mean and median for a month in Figure R2 and it shows clearly that any differences are small.

[Figure]

Figure R2: Comparison of median and mean differences of MH in a month.

d) Assuming these data are from some reanalysis dataset, one must be careful about interpreting them for smaller MLH values (higher uncertainty).

**RESPONSE:** We accept this point. The data show greater variability (expressed as RSD) at lower values of MLH, reflected in Table R1 below, and Figure R3. We have commented upon this in the manuscript (L403).

"TPN showed a negative exponential dependence upon MLH, which became more scattered at lower values of MLH, probably reflecting the larger relative errors in MLH estimates at smaller values."

Table R1: The mean and standard deviation of TPN for four ranges of MLH.

| Stat. of TPN for all data | Mixing Height | | | |
|---|---|---|---|---|
| | <500 | 500-1000 | 1000-1500 | >1500 |
| Mean | 36181 | 23667 | 17046 | 18301 |
| StanDev | 15158 | 8127 | 5972 | 6671 |
| %RSD | 42 | 34 | 35 | 36 |

[Figure]

Figure R3. Variations of Total Particle Number (TPN) as a function of mixing height (MLH) for all data and for each season.

2. Connected to the previous point, interpreting the diurnal patterns of particle number concentrations (e.g., in the abstract: page 2, line 58) only in terms of traffic levels can be misleading. In addition to MLH, cooking in all seasons, biomass burning for heating in autumn/winter, and agricultural burning in autumn can all emit particles in the accumulation mode. I suggest considering these processes and additional (non-traffic) sources carefully throughout the manuscript.

**RESPONSE:** We are pleased to accept this point, and have amended the abstract (L61) as follows: "The diurnal pattern of particle numbers is strongly reflective of a road traffic influence upon concentrations, especially in autumn and winter, although other sources such as cooking and domestic heating may influence the evening peak."

Also, the following has been added to Section 3.2 (L330): "While road traffic clearly influences the diurnal pattern in PN, other sources including cooking and domestic combustion are likely to contribute."

Minor comments:

3. (Page 3, line 82) "Although the sources of particles are mostly local (Hama et al., 2020)…" This citation may be misread as a reference from Delhi given the discussion before. Furthermore, the authors should be careful and intentional in writing "particles" (number or mass?).

**RESPONSE:** We have amended the text as follows to reflect this point.

"….Although the overall emission sources in India are dominated by traffic, industry, construction, and local biomass burning, haze pollution events in Delhi are frequently related to the large-scale open burning of post-harvest crop residues/wood during the crop burning season in nearby rural regions (Cusworth et al. 2018; Bikkina et al. 2019; Kanawade et al., 2020). Furthermore, meteorological factors play an important role in influencing concentrations of air pollution (Tiwari et al., 2014; Yadav et al., 2016; Guo et al., 2017; Dumka et al. 2019; Kumar et al. 2020)."

4. (Page 5, line 128) Is this site "representative of an urban background environment"? I understand that using such terminologies will always be difficult in a polluted megacity like Delhi with large spatial variations in air pollution levels. However, this site seems to be close to a major arterial road (please add this distance in the site description) and given the importance of traffic-emissions in particle number concentrations, "urban background" may not be a correct representation.

**RESPONSE:** The following text (L133) has been added as clarification.

"…The sampling location was ~15 m above ground level on the 5th floor of the Civil Engineering Department at the Indian Institute of Technology Delhi (IIT Delhi) campus, located in New Delhi, representative of an typical urban background environment in a Megacity (28.545 N, 77.193 E) (Figure S1).The measurement station is 120 m distant from a major arterial road. "

5. (Page 10, line 255) The difference in distance from the major arterial road between the two studies could be another factor in differences in concentrations observed (same comment for page 14, line 373-375). Also, in line 256, do you mean "sampling period" instead of "sampling time"?

**RESPONSE:** Gani et al. (2019) stated the measurement location as "we installed a suite of online aerosol measurement instrumentation at the Indian Institute of Technology Delhi (IITD) campus in South Delhi. The instruments are situated in a temperature-controlled laboratory on the top floor of a four-story building. The nearest source of local emissions is an arterial road located 150 m away from the building." The two sites Gani et al. and ours are ~ 30m away from each other. The Gani et al. site is about 150 m away from the road and on the top of a four storey building. Our study is 120 m away from the road and on the top of a five storey building.

"sampling period" is more appropriate. It has been changed.

6. I am unable to interpret the colors/legend in Figure 5.

**RESPONSE:** The figure 5 has been recreated.

Technical corrections:

7. (Page 16, line 430) Cited article is missing from the bibliography.

**RESPONSE:** The reference has been added the list. It is: "Thamban, N.M., Lalchandani, V., Kumar, V., Mishra, S., Bhattu, D., Slowik, J.G., Prevot, A.S.H., Satish, R., Rastogi, N., Tripathi, S.N.: Evolution of size and composition of fine particulate matter in the Delhi megacity during later winter, Atmospheric Environment, 267,118752, doi.org/10.1016/j.atmosenv.2021.118752. 2021."

8. Recent updates to ACP/Copernicus guidelines state that "it is important that the colour schemes used in your maps and charts allow readers with colour vision deficiencies to correctly interpret your findings." This means that jet/rainbow (non-uniform) color scales need to be changed to other appropriate color scales. Please update figures 4, 11, S5, and S12 accordingly. More here: https://www.atmospheric-chemistry-and-physics.net/submission.html#figurestables

**RESPONSE:** We have used colours commonly found in journal papers, and would prefer not to re-draw.

**References:**

Gani, S., Bhandari, S., Seraj, S., Wang, D. S., Patel, K., Soni, P., Arub, Z., Habib, G., Hildebrandt Ruiz, L., and Apte, J. S.: Submicron aerosol composition in the world's most polluted megacity: the Delhi Aerosol Supersite study, 19, 6843–6859, https://doi.org/10.5194/acp-19-6843-2019, 2019.

**RESPONSE:** The paper was published in ACP as Gani et al. 2020, which is the correct date.

**REVIEWER #2**

The MS as a measurement report mainly deals with deriving wide-range particle number size distributions from a mobility particle size spectrometer and a particle optical counter, and also presents its application, results and conclusions for Delhi. Its basic ideas, topic and outcomes are timely, important and of interest for the research community. Its evaluation methods are mostly plausible. At the same time, the MS contains several conceptional errors, which are severe, and are misleading in many ways. They cannot be tolerated in a journal with an impact factor of 6.546 and should be definitely corrected or removed before any reliable evaluation of the MS could be finalised. There is also a large number of smaller discrepancies in the text and figures which should be improved and handled. The list below gives only examples of them.

Major comments

1. The authors may want to emphasize better the need for and advantages of wide-range particle number size distributions.

**RESPONSE:** We have also described the importance of wide-range particle number size distribution in the MS, in the Introduction, lines 114-121 as below:

"The wide range PNSD is important to describe all sources of inhalable particles (<10 μm). It is not easy to separately identify particles arising from resuspension, sea salt and construction, or from

brake wear and combustion or vehicle exhaust, using only the <0.5 µm particle size range. Harrison et al. 2011 reported that using wide range particle sizes in source apportionment was extremely successful in identifying the separate contributions of on-road emission including brake wear and resuspension. Although there are a few studies of wide range particle characterization in Beijing (Jing et al., 2014) and source apportionment in Venice, Italy (Masiol et al., 2016), there has been no previous wide range PNSD study in Delhi. In this study….."

2. The abbreviation PM expresses particulate matter, thus the set of aerosol particles (and not their any property). It is used over the whole MS in a fuzzy manner, sometimes hinting at the PM mass. A very clear distinction between the PM mass and particle numbers should be made throughout the text since the main objectives of the MS are related to particle number concentrations and size distributions. This seemingly small discrepancy results in several misleading sentences and erroneous formulations. Examples are L90-92 and L95-96: with regard to the mass or particle number? With respect to this, PM10 volume is correct, while the formulation "PN10 number" is unusual, not consequent and, therefore, confusing (L50).
**RESPONSE:** We have revised the manuscript to make this clearer. We retain the term PN10 to describe the particle number below 10µm diameter, but have defined it clearly.

L93-99 in the MS has been amended as follows: …"Mostly they have reported that biomass burning contributes greatly to $PM_{2.5}$ mass while traffic contributes heavily to $PM_{10}$ mass in Delhi. Residential energy use contributes 50 % of the $PM_{2.5}$ mass concentration and the construction sectors are also considered an important source of particle mass (Guttikunda et al., 2014; Butt et al., 2016; Conibear et al., 2018). Furthermore, it is particularly important to understand the absolute contribution and sources of different sizes of particles within $PM_{2.5}$. A recently published paper by Das et al. (2021) highlighted that <250 nm particles contribute a significant proportion of the total $PM_{2.5}$ mass and are a potentially important link with human health."

We also added some detail in L50 as below:
L50: The ultrafine fraction (15-100 nm) accounts for about 52 % of all particles by number ($PN_{10}$- total particle number from 15 nm to 10 µm), but just 1 % by $PM_{10}$ volume ($PV_{10}$- total particle volume from 15 nm to 10 µm).

3. The situation is somewhat similar with the particle size distribution (L98). The expression is meaningless in its present form. Size distributions are exclusively related to properties of particles, which are missing from the expression. There are mass size distributions, surface area size distributions, particle number size distributions, etc. This should be added at many places in the MS and the title should also be changed accordingly.
**RESPONSE:** In L102, the PSD refers to the number size distribution, so we added "number" in the sentence. We reviewed the whole paper and changed PSD to PNSD when it referred to particle number.
"The Particle Number Size Distribution (PNSD)….."
We do not think it useful to amend the title of the paper as we interpret size distributions expressed in a variety of ways.

4. The size distributions consist of modes (peaks which extend from -∞ to +∞). Their significant content or area can be approximated by a size range. The modes are: nucleation, Aitken, accumulation and coarse, while the classical size fractions are ultrafine, fine and coarse or PM10, PM2.5, PM1.0 and etc. The size fractions can be defined freely. The authors should not, however, mix the modes and the size fractions. Accumulation size fraction could be preferred to accumulation mode in this aspect. In addition, in L54-L55: which mode?; in L380: area distribution?
**RESPONSE:** The manuscript has been revised for greater clarity.
In L427 the figure number is wrong. "Figure 5" was changed to "Figure 6".

5. The authors are requested to discuss the dependency of the particle density on the size (L232), to describe how the particle volume was actually calculated from particle number and particularly, what the resulting uncertainties from the conversion were (L380), and specify the method of deriving GMDs (L409+).

**RESPONSE:** The following text has been added to the Methods section.
The dependence of particle density on the size (L244-246):
Estimation of particle density as a function of size is extremely difficult, and there are few data for particle density from Delhi. Since Gani et al (2020) used the density of PM at the same location as in our study, we used the same density value to convert PN to PM mass.

The methods of deriving GMDs and GRs (L252-261):
The cumulative frequency of observations as a function of particle size was calculated for each hour of the day. Standard central measures from the cumulative frequency plots were represented by the geometric mean diameter (GMD) for each size distribution. They were used to examine particle growth processes. Firstly, the growth of GMD was estimated visually from the diurnal GMD data plot (Fig. 8). The minimum growth time used for estimation of the growth rate (GR) was selected as three hours, and if the growth lasted for long enough, the GR was estimated. The observed growth of the GMD of the particle was quantified by fitting the GMD of particles during the growth process event over a period of time 't' (eq.1). Detailed information on the method can be found in Sarangi et al. (2015; 2018).
Growth rates (nm/hour)-GR =dGMD/dt                              (1)

6. The diurnal variation of BC (Fig. 2) and the increasing part of the particle number size distributions with decreasing particle diameter in the range <ca. 20 nm (Fig.6 ) could also be explained and discussed (better).
**RESPONSE:** We have given careful thought to this. It is not clear what feature of Figure 6 the reviewer is drawing attention to. Figure 7 shows diurnal features, but the <20 nm part of the size distribution does not show any obvious dependence upon time-of-day. The diurnal variation in BC is discussed in the last paragraphs of both Section 3.4 and Section 3.5.

Minor comments
 7. Abbreviations TEOM-FDMS (L136-137), HOA, SVOOA, BBOA and LVOOA (L431) are not resolved, UFP is defined several times (L229, L262, L368), SMPS is explained not at its first appearance in the body text (L146 vs. 158).
**RESPONSE:** All have been corrected in the revised MS.

8. The size distribution and other properties obtained by merging the SMPS and Grimm optical spectrometer data are better to be called SMPS-OP data instead of SMPS-Grimm data, with OP standing for particle optical counter (L164, L178, L199+).
**RESPONSE:** We are pleased to adopt this suggestion and have amended accordingly.

9. Part L201-L206 belongs more to the description of the experimental methods than to section Data and Quality Management. Part L220-L230 contains extensive repetitions, and its rest should be shifted to methodology.
**RESPONSE:** The entire Methods section has been edited to remove repetition and improve clarity.

10. The authors should revisit their rounding off strategy. Example: instead of 36730 cm$^{-3}$ give $3.7 \times 10^3$ cm$^{-3}$ (L244).
**RESPONSE:** We accept this point, but prefer to use 4 significant figures, hence $36.73 \times 10^3$ cm$^{-3}$.

---

## Author Response (AR2)

MS No.: acp-2021-860 MS type: Measurement report
Title: Measurement Report: Interpretation of Wide Range Particulate Matter Size Distributions in Delhi
Author(s): Ülkü Alver Şahin et al.

**RESPONSE TO REVIEWERS (2)**

**REPORT #1**

I appreciate the edits made by the authors throughout the manuscript based on initial set of reviews.

However, I find the response of the authors to not change their current colorscale (jet/rainbow) unsatisfactory.

Authors: "We have used colours commonly found in journal papers, and would prefer not to redraw."

I understand that the aerosol community has used the jet/rainbow colorscale for decades and it is inconvenient for us to change from the "standard" rainbow/jet colorscale. But, given the guidelines, I have to point this out. The Copernicus guidelines (https://www.atmospheric-chemistry-and-physics.net/submission.html#figurestables) clearly state: "For more information on the background and importance of addressing this issue, we refer to Stoelzle & Stein (2021) (https://hess.copernicus.org/articles/25/4549/2021/)." The cited article literally discusses why "rainbow colormap" (color scheme in question) is misleading.

As such, I reiterate, the authors should change the colorscales in figures 4 and 11, S5, S8, and S13 to a uniform colorscale (https://www.nature.com/articles/s41467-020-19160-7)

While it is ultimately the editor's call, I want to note that I have made this comment on other manuscripts — and want to be consistent — where the authors changed to a non-misleading (uniform) colorscale.

RESPONSE: The authors have amended the colorscale in the figures identified by the reviewer in the way requested.

.

[revised manuscript text omitted]

**Table S1:** Merged Data Grade and Set limits . [O: Is there an overlap? N: Is the data scattered over the overlap? F: Fraction of points which match, S: Smoothness of overlap across the SMPS and GRIMM data.]

| Criterion | | | | Point |
|---|---|---|---|---|
| O:
1: Yes;
0: No. | N:
– None;
2- either SMPS or GRIMM;
– over both SMPS and GRIMM. | F:
– All;
2- most;
1- None | S:
– smooth;
2- bumpy;
1- Stepped. | $(9/27) \times O \times N \times F \times S$

is perfect, 0 and 1 are unacceptable and are excluded from data set. |
| 1 | 3 | 3 | 3 | 9 (PERFECT) |
| 1 | 3 | 3 | 2 | 6 (KEEP) |
| 1 | 3 | 2 | 3 | 6 (KEEP) |
| 1 | 2 | 3 | 3 | 6 (KEEP) |
| 1 | 3 | 1 | 2 | 6 (ACCEPTABLE) |
| 1 | 3 | 1 | 3 | 6 (ACCEPTABLE) |
| 1 | 2 | 1 | 3 | 6 (ACCEPTABLE) |
| 1 | 1 | 2 | 3 | 2 (IMPROVE) |
| 1 | 1 | 3 | 2 | 2 (IMPROVE) |
| 1 | 3 | 2 | 1 | 2 (IMPROVE) |
| 1 | 1 | 1 | 3 | 1 (REJECT) |
| 1 | 1 | 3 | 1 | 1 (REJECT) |
| 1 | 3 | 1 | 1 | 1(REJECT) |
| 0 | 3 | 3 | 3 | 0 (REJECT) |

[Figure]

**Figure S2:** Time series of particle number counts for the sum of nucleation, Aitken, accumulation, fine and coarse modes of PM from the SMPS and Grimm instruments for all season in Delhi.

[Figure]

**Figure S3:** Particle concentration change during Diwali in 2018 in Delhi.

[Figure]

**Figure S4:** Comparison of PM$_{2.5}$ measured by SMPS+GRIMM and TEOM in Delhi for three seasons.

[Figure]

**Figure S5:** Comparison of PM$_{2.5}$ measured by SMPS+GRIMM and TEOM with relative humidity in Delhi for three seasons.

**Table S2:** Descriptive statistics of particle number counts (#/cm$^3$) calculated using every 1 hour measurements for nucleation, Aitken, accumulation, large fine and coarse modes of PN between 15 nm and 10 µm derived by SMPS and Grimm in all seasons.

| Seasons | PN modes | Descriptive Statistics | | | |
|---|---|---|---|---|---|
| | | Range (Min-Max) | Mean | Median | Std. Deviation |
| Autumn | Nucleation | 338-5033 | 1296 | 1147 | 617 |
| | Aitken | 2406-44009 | 12828 | 11416 | 6429 |
| | Accumulation | 4620-46655 | 15186 | 15191 | 5677 |
| | Large Fine | 1-126 | 9 | 6 | 13 |
| | Coarse | 0-5 | 1 | 1 | 1 |
| | Total | 7507-62756 | 29355 | 28528 | 9883 |
| Summer | Nucleation | 302-2504 | 821 | 780 | 310 |
| | Aitken | 2237-32521 | 9965 | 7963 | 6084 |
| | Accumulation | 2871-27211 | 8107 | 5899 | 4736 |
| | Large Fine | 1-582 | 11 | 6 | 9 |
| | Coarse | 0-5 | 2 | 2 | 1 |
| | Total | 5436-57565 | 18906 | 15362 | 10121 |
| Winter | Nucleation | 510-3159 | 1489 | 1430 | 541 |
| | Aitken | 3356-51293 | 17610 | 15571 | 9682 |
| | Accumulation | 3901-50466 | 17599 | 15221 | 9276 |
| | Large Fine | 3-602 | 12 | 10 | 8 |
| | Coarse | 0-9 | 2 | 1 | 1 |
| | Total | 11506-95068 | 36730 | 33053 | 17815 |

[Figure]

[Figure]

**Figure S6:** Normalized time variation of all particle fractions (Nuc: Nucleation <25 nm, Ait: Aitken 25-100 nm, Acc: Accumulation 100-1000 nm, Fine: 1-2.5 µm, Coarse: 2.5-10 µm number counts (between 15.1 nm and 10 µm) derived from the SMPS and Grimm instruments for winter, autumn and summer in Delhi.

[Figure]

**Figure S7:** Average diurnal variation of meteorological parameters during the PN measurement campaign in autumn, summer and winter.

[Figure]

**Figure S8:** Variations of Total Particle Number (TPN) as a function of ventilation coefficient (VC). a) and b): Each scatter point is a hourly average and is colour coded by hour and month. c): Each scatter point is a daily average and is colour coded by month. Note that the summer data is not enough to assess.

[Figure]

**Figure S9:** Hourly idling traffic at Lodhi Road signalized intersection in Delhi (data from Dhyani et al. 2019).

**Table S3:** Day and night summary of the wide range particle sizes derived from the merged data. Mean and standard deviation calculated by hourly data. The modes are based on SMPS and GRIMM observations. Nucleation = 10 - 25 nm, Aitken = 25-100 nm, Accumulation = 100-1000 nm, Large Fine = 1000-2500 nm, Coarse = 2500-10000 nm. UFP = Nucleation +Aitken, PN1 = UFP+Accumulation, PN10= PN1+Large Fine+Coarse, N/D = the ratio of Night to Day.

| | Winter | | | Autumn | | | Summer | | |
|---|---|---|---|---|---|---|---|---|---|
| | Day (N=252) | Night (N=284) | N/D | Day | Night | N/D | Day (105) | Night (111) | N/D |
| Number Concentration from GRIMM+SMPS | | | | | | | | | |
| Nucleation (#x$10^3$ cm$^{-3}$) | 1.3±0.5 | 1.6±0.5 | 1.3 | 1.2±0.5 | 1.3±0.7 | 1.1 | 0.8±0.4 | 0.8±0.2 | 1.0 |
| Aitken (#x$10^3$ cm$^{-3}$) | 13.5±6.7 | 21.3±10.4 | 1.7 | 10.7±5.4 | 13.1±6.2 | 1.2 | 8.3±4.7 | 11.5±6.8 | 1.4 |
| Accumulation (#x$10^3$ cm$^{-3}$) | 11.8±5.9 | 22.7±8.7 | 1.9 | 12.1±4.7 | 17.8±3.8 | 1.5 | 6.5 ±3.6 | 9.6±5.1 | 1.5 |
| Large Fine (# cm$^{-3}$) | 10.1±8.6 | 13.0±7.6 | 1.3 | 6.1±5.8 | 8.5±7.3 | 1.4 | 7.7±5.9 | 13.1±11.3 | 1.7 |
| Coarse (# cm$^{-3}$) | 1.6±1.3 | 1.8±1.2 | 1.1 | 1.1±0.6 | 1.1±0.6 | 0.9 | 1.9±0.9 | 2.2±1.2 | 1.2 |
| UFP (#x$10^3$ cm$^{-3}$) | 14.7±7.6 | 22.9±11.3 | 1.6 | 13.0±6.5 | 15.1±7.7 | 1.1 | 9.1±5.5 | 12.3±7.0 | 1.3 |
| PN1 (#x$10^3$ cm$^{-3}$) | 26.5± 13.5 | 45.6±20.0 | 1.7 | 25.0±11.1 | 33.3±12.5 | 1.3 | 15.6±9.1 | 21.9±12.1 | 1.4 |
| PN10 (#x$10^3$ cm$^{-3}$) | 38.1±11.8 | 60.4±28.8 | 1.6 | 32.2±18.2 | 45.2±29.8 | 1.4 | 25.2±15.9 | 37.2±24.6 | 1.5 |
| Mass Concentration | | | | | | | | | |
| Grimm+Smps-PM$_1$ (μg m$^{-3}$) | 202.7±103.3 | 340.3±109.3 | 1.7 | 259.5±151.8 | 357.7±182.8 | 1.4 | 84.8±47.5 | 121.8±62.3 | 1.4 |

| | | | | | | | | |
|---|---|---|---|---|---|---|---|---|
| Grimm+Smps PM$_{2.5}$ (μg m$^{-3}$) | 231.8±113.5 | 375.3±117.7 | 1.6 | 277.2±156.5 | 382.4±198.5 | 1.4 | 106.4±55.5 | 151.5±76.5 | 1.4 |
| Grimm+Smps PM$_{10}$ (μg m$^{-3}$) | 377.3±161.5 | 532.5±175.9 | 1.4 | 362.1±159.5 | 455.3±198.1 | 1.2 | 260.2±122.6 | 331.8±171.8 | 1.3 |
| Teom-PM$_{2.5}$ (μg m$^{-3}$) | 158.0±78.9 | 196.3±84.0 | 1.2 | 214.8±121.3 | 228.5±112.5 | 1.0 | 117.2±53.8 | 123.7±56.2 | 1.0 |
| BC (μg m$^{-3}$) | 5.0±4.0 | 14.8±11.7 | 2.9 | 11.4±7.8 | 20.1±9.0 | 1.8 | 11.2±9.5 | 5.2±3.5 | 2.1 |
| NO (ppb) | 17.6±42.4 | 103.9±99.8 | 6.1 | 32.9±58.5 | 123.8±85.8 | 3.7 | 4.4±12.7 | 107.1±127.8 | 26.5 |
| NO2 (ppb) | 59.5±28.0 | 76.7±25.5 | 1.3 | 88.2±44.0 | 69.5±21.0 | 0.8 | 40.3±34.0 | 87.2±49.4 | 2.1 |

[Figure]

**Figure S10:** Hourly average particle number (a), volume (b) and area (c) distributions derived from the SMPS and Grimm instruments in autumn, summer and winter in Delhi. (Between 15 nm and 10000nm).

[Figure]

**Figure S11:** Average PSD for each hour and season in Delhi.

[Figure]

**Figure S12:** Correlation coefficients (x100) of PN within size fractions and NO, NO₂, BC.

[Figure]

[Figure]

**Figure S13:** Polar plots of BC (top panels) NO, NO$_2$ (bottom panels) in winter (left), autumn (midle) and summer (right) in Delhi.